# Circulating Cell-Free Nucleic Acids as Biomarkers for Diagnosis and Prognosis of Pancreatic Cancer

**DOI:** 10.3390/biomedicines11041069

**Published:** 2023-04-01

**Authors:** Anelis Maria Marin, Heloisa Bruna Soligo Sanchuki, Guilherme Naccache Namur, Miyuki Uno, Dalila Luciola Zanette, Mateus Nóbrega Aoki

**Affiliations:** 1Laboratory for Applied Science and Technology in Health, Carlos Chagas Institute, Oswaldo Cruz Foundation (Fiocruz), Prof Algacyr Munhoz Mader 3775 Street, Curitiba 81350-010, Brazil; 2Center for Translational Research in Oncology (LIM24), Departamento de Radiologia e Oncologia, Instituto do Câncer do Estado de São Paulo (ICESP), Hospital das Clínicas da Faculdade de Medicina da Universidade de São Paulo (HCFMUSP), São Paulo 01246-000, Brazil

**Keywords:** pancreatic cancer, circulating cell-free nucleic acids, cell-free DNA, diagnostic, prognostic

## Abstract

A lack of reliable early diagnostic tools represents a major challenge in the management of pancreatic cancer (PCa), as the disease is often only identified after it reaches an advanced stage. This highlights the urgent need to identify biomarkers that can be used for the early detection, staging, treatment monitoring, and prognosis of PCa. A novel approach called liquid biopsy has emerged in recent years, which is a less- or non-invasive procedure since it focuses on plasmatic biomarkers such as DNA and RNA. In the blood of patients with cancer, circulating tumor cells (CTCs) and cell-free nucleic acids (cfNAs) have been identified such as DNA, mRNA, and non-coding RNA (miRNA and lncRNA). The presence of these molecules encouraged researchers to investigate their potential as biomarkers. In this article, we focused on circulating cfNAs as plasmatic biomarkers of PCa and analyzed their advantages compared to traditional biopsy methods.

## 1. Introduction

Pancreatic cancer (PCa) is the third leading cause of cancer deaths in the United States [1] and is projected to become the second leading cause by 2030 [2]. This increase is largely due to its high lethality, which remains to be at 90% even as the five-year survival has nearly doubled in the last two decades [1], due mainly to improvements in perioperative systemic treatments [3,4]. Pancreatic ductal adenocarcinoma (PDAC) is by far the most common subtype of PCa, accounting for more than 85% of cases [5]. At initial presentation, half of the patients already have a metastatic disease [6], and 20–30% of patients have locally advanced tumors with major vascular invasion, which precludes resection, with only 15–20% of patients with PDAC considered to be candidates for surgical resection [7]. As there is no indication for general population screening, the vast majority of patients are diagnosed after symptomatic presentation. Initial symptoms are usually vague and nonspecific, such as indigestion, fatigue, loss of appetite, and weight loss. Jaundice, which is usually associated with PCa, is an initial symptom in only 12% of patients, even though nearly 50% will develop it in the course of their illness [8]. New-onset diabetes may be the first clinical evidence of PDAC. At 13 to 18 months before the diagnosis of PDAC, patients may develop hyperglycemia and, paradoxically, may present weight loss, which may start 18 months before diagnosis [9]. PDAC-related new on-onset diabetes is usually more severe and affects older patients, therefore, Sharma et al. developed a risk score based on age, weight, and blood glucose changes, to assist selecting high risk patients for PCa screening programs [10], and there is an ongoing randomized trial using that score as an automated algorithm in a healthcare database [11]. 

Several behavioral factors are correlated with pancreatic cancer incidence, such as tobacco, diet, alcohol, and body mass index. In this context, tobacco is a well-known and consolidated risk factor for pancreatic cancer [12,13,14], with increased risk correlated with cigarette number usage, where smoking 35 or more cigarettes per day shows an odds ratio for pancreatic cancer of 3.0 (95% CI 2.2–4.1) [15]. Interestingly, quitting smoking (former smokers) reduces pancreatic cancer risk with increasing years after cessation, and 10–20 years after smoking cessation the risk of pancreatic cancer for former smokers is similar to that for never smokers [13,14]. Body mass index is a well-known and directly correlated risk factor for pancreatic cancer [16]. It has been demonstrated that overweight (BMI 25–29.9 kg/m^2^), obesity (BMI 30–34.9 kg/m^2^), and severe obesity (BMI > 35 kg/m^2^) represent odds ratios of 1.19 (95% CI 1.02–1.40), 1.25 (95% CI 1.02–1.55), and 1.62 (95% CI 1.19–2.21) compared to individuals with normal weight [17]. Alcohol is not a consensus risk factor for pancreatic cancer; however, evidence has demonstrated that heavy alcohol consumption represents an elevated odds ratio for its incidence. The Pancreatic Cancer Case Control Consortium (PanC4), NIH-AARP Diet and Health and American Cancer Society Cancer Prevention Study II showed an odds ratio of (OR 1.6, 95% CI 1.2–2.2), f 1.45 (95% CI 1.17–1.80), and 1.32 (95% CI 1.10–1.57) for heavy drinkers compared to light or non-drinkers [18,19,20].

Carbohydrate antigen 19-9 (CA19-9) is the only widely used biomarker for PCa. It is a sialytated Lewis blood group antigen and about 5% of the population does not express it [21]. The sensitivity for a diagnosis of PAc is 79% and specificity 82% [22]. Although CA19-9 could be elevated in a wide variety of benign diseases, especially benign bile duct obstructions [23], usually the relief of biliary obstruction and normalization of bilirubin is associated with decline, if not normalization, of CA19-9 levels [24]. CA19-9 is useful to predict resectability, survival, and it aids physicians through patients’ clinical follow-ups. Patients with CA19-9 lower than 37 U/mL have resectability of 79.7% and 27.2% shaould be alive after 5 years; on the contrary, patients with CA19-9 above 1000 U/mL have resectability of less than 50% and 5-year survival is null [25]. CA19-9 is also very important to evaluate a response to neoadjuvant treatment, which is increasingly frequent, since a drop in CA19-9 levels is the single most important marker of clinical response and may predict resectability and good prognosis [26].

Imaging tests are pivotally important to manage PDAC. Computed tomography (CT) and magnetic resonance imaging (MRI) can both be used through the onset of the disease in order to establish diagnosis and staging. CT and MRI have very similar sensitivity (89% for both) and specificity (90 an 89% respectively) regarding the diagnosis of PDAC, and also perform very similarly regarding vascular invasion, which is of paramount importance to determine resectability [27]. At diagnosis, PDAC can be classified as resectable, unresectable or locally advanced, metastatic, and borderline resectable, according to the anatomic findings at initial work up. In summary, resectable tumors are those without involvement of vascular structures such as celiac axis or superior mesenteric artery (SMA); unresectable tumors are those with arterial encasement; and borderline resectable disease is characterized as potentially resectable but with high risk of positive margins because of arterial proximity or minor venous abutment. However, the definitions may vary from every other classification [28]. Although CT and MRI present equivalent performance for diagnosis of PDAC and evaluation of vascular involvement, MRI is more sensitive for PAc liver metastasis detection [29]. 

Fluorodeoxyglucose (FDG) positron emission tomography (PET/CT), which is a functional imaging test that evaluates glucose metabolism, may be useful to distinguish PCA from benign inflammatory diseases such as autoimmune pancreatitis, characterized as diffuse uptake of FDG [30]. PET/CT has a poorer anatomical definition compared to iodine contrast-enhanced CT, with similar sensitivity for a diagnosis of pancreatic cancer, however, it has superior specificity regarding metastatic status [27]. Although it should not be considered to be a routine test for PDAC, FDG-PET/CT could be an important tool to predict and to assess response to neoadjuvant chemotherapy, as there is a correlation between a drop in FDG uptake and histopathologic tumor regression [31]. 

Endoscopic ultrasound (EUS) allows direct visualization of pancreatic parenchyma, however, it is not superior to tomography to evaluate vascular invasion or lymph nodal status, and it should not be part of routine evaluation for resectable PDAC [27,32]. EUS is the gold standard method for pancreatic tissue acquisition, usually through fine needle aspiration (FNA), because it is related to lower frequency of peritoneal carcinomatosis than percutaneous biopsy [33]. The FNA sensitivity and specificity for PDAC are 90.8% and 96.5%, respectively [34]. Currently, neoadjuvant chemotherapy has become the standard treatment for borderline resectable tumors and will probably overcome straightforward surgery for resectable disease in the coming years [35]; therefore, efforts in order to improve diagnosis are important because histological confirmation is indispensable for initiating chemotherapy.

Unlike others, such as breast cancer, pancreatic cancer does not display a very long latency for the development of metastatic disease and is primarily considered to be a metastatic disease on clinical parameters, where only 10–15% of patients present resectable disease, with the vast majority of patients presenting as locally advanced and systematic metastatic, especially in the liver and lungs [36,37,38]. In this disease, premalignant lesions (carcinoma in situ) are considered to be precursors of invasive and metastatic adenocarcinoma, termed pancreatic intraepithelial neoplasia (PanINs) and defined as microscopic, non-invasive epithelial neoplasm of the pancreatic duct system. PanINs are classified according to a four-tier classification system, as PanIN-1A, PanIN-1B (low-grade PanINs), PanIN-2 (intermediate grade Pan-INs), and PanIN-3 (high-grade PanIN), reflecting a progressive increase in histologic grade culminating in invasive and metastatic neoplasia [39,40,41,42].

## 2. Pancreatic Cancer Genome

### 2.1. Germline Context

Despite PCa emerging in a multifactorial context, this review does not focus on behavioral risk factors such as obesity, smoking, or alcohol. Our goal is to discuss the general genetic background, especially circulating cell-free nucleic acids (cfNAs) as plasmatic biomarkers for PCa diagnosis and prognosis. The genetic context of PCa is relatively well described, and genomic evidence suggests a genetically heterogeneous disease with different molecular signatures [43,44]. Unlike other cancers, only a small percentage (5–10%) of PCa cases are linked to heredity and connected with the germline genetic background [45,46,47,48,49]. Familial PCa is characterized by a family having two or more cases among first-degree relatives without any other hereditary cancer syndrome being observed [50]. The risk of developing PCa in families affected by familial PCa increases depending on the number of affected members, varying from a 4.6-fold increase in families with two PCa cases to 32-fold for families with three PCa cases [49,51].

In the context of hereditary PCa, the most frequent germline alterations rely on *CDKN2A*, *TP53*, MLH1, BRCA1, BRCA2, and ATM, of which the former two will be discussed. Importantly, BRCA genes are the most consolidated genetic biomarkers for breast cancer and are becoming important genetic biomarkers for PCa as well, accounting for up to half of all germline mutations found [52,53,54]. Germline mutation incidence in unselected PCa patient cohorts has been shown to range from 0.3 to 2.3% for BRCA1 and from 0.7 to 6% for BRCA2 [53,55,56,57,58,59,60]. This wide variation relies on several factors, including cohort composition, the number of patients with family histories of cancer, and patient age, which highlights the genetic heterogeneity of PCa [56]. BRCA genes are part of the DNA double-strand break repair machinery [61], and their mutations increase genomic instability through faulty homologous recombination at stalled replication forks, thus, increasing the rate at which somatic mutations occur [62].

In the germline genetics of PCa, single nucleotide polymorphism (SNP) variations are logical and targeted as risk and protective factors. Genome-wide association studies (GWASs) have been conducted for almost 15 years to determine germline genetic variations that may be involved in PCa. In 2009, high-throughput SNP results were reported in the scientific literature. For example, McWilliams [63] studied 1143 patients with pancreatic adenocarcinoma and 1097 healthy controls, focusing on 28 genes known to be directly and indirectly involved in double-stranded break repair, as well as PRSS1, PRSS2, and *CDKN2A*, using a 768 SNP panel, and showed no significant association of any genes with altered PCa risk. However, further results demonstrated several genetic loci involved with PCa. The first such study was published in 2009, initially with 1896 cases and 1939 controls and a replication phase with 2457 cases and 2654 controls, and showed an association between a locus on 9q34 and PCa marked by the SNP *rs505922* (OR 1.20), mapped on the first intron of the ABO blood group gene, which shed light on the genetic epidemiology of PCa [64]. This report led to further exploration of blood group alleles and the risk of PCa [65,66]. Further GWASs have demonstrated several SNPs involved with PCa risk, representing a wide variety of cohort constitutions and methodologies with contrasting results. One subsequent GWAS was performed with 3851 PCa individuals and 3934 controls, identifying eight SNPs that map to three loci on chromosomes 13q22.1, 1q32.1, and 5p15.33 [67]. In 2014, the data from another GWAS that included 7683 individuals with PCa and 14,397 controls of European descendants, identified several new genetic regions associated with PCa risk, together with novel candidate genes implicated in pancreas development, pancreatic beta-cell function, and predisposition to diabetes [68]. Zhang et al. performed another GWAS based on the 1000 Genomes (1000 G) Project data and association analysis using 5107 cases and 8845 controls, combined with a two-staged replication in an additional 6076 cases and 7555 controls. They found three new Pca risk SNPs (*rs2816938* at chromosome 1q32.1, *rs10094872* at 8q24.21, and *rs35226131* at 5p15.33) in addition to variants at 13 chromosomal loci linked with Pca susceptibility previously reported in individuals of European descent [69]. Subsequently, the largest Pca GWAS so far that included 9040 patients and 12,496 controls of European ancestry, identified five new susceptibility loci for PCa risk [70]. Moreover, a recessive genetic model represents a promising tool for identifying additional risk variants for PCa, as demonstrated by re-analyzing the largest GWAS (PanScan) and the Pancreatic Cancer Case-Control Consortium (PanC4), including 8769 cases and 7055 controls of European ancestry. In this report, six additional SNPs were risk associated according to a recessive inheritance model finding that was replicated in a large cohort of 3212 cases and 3470 controls collected from the PANcreatic Disease ReseArch (PANDoRA) consortium. Although none of the six SNPs reached the conventional threshold for genome-wide significance, the authors found three loci with specific recessive effects compared with the additive effects: *rs4626538* (7q32.2), *rs7008921* (8p23.2), and *rs147904962* (17q21.31) [71].

More recently, germline genotyping for patients has been moving towards more assertive cohorts related to ethnicity, age, and the molecular and clinical characteristics of tumors [72,73,74,75]. The composition of the cohorts chosen, however, may bias the molecular epidemiology findings: A recent report using 2039 patients and 32,592 controls in the Japanese population identified three genome-wide significant loci (13q12.2, 13q22.1, and 16p12.3), among which 16p12.3 has not been reported in the Western population [76]. SNPs located in non-coding regions are also emerging as important players, as recently reported by Corradi and colleagues (2021), who analyzed 9893 PDAC cases and 9969 controls and reported a significant correlation between *rs7046076* SNP and the risk of developing PDAC. This SNP is located in the NONHSAG053086.2 (lnc-SMC2-1) gene, and this polymorphism is predicted to interfere and disrupt the binding of this lncRNA with hsa-mir-1256, a molecule that is correlated with regulation of genes involved in cell cycles, such as CDKN2B [77]. A re-analysis of data from two consortia (PanGenEU and PANDoRA) with 14,062 PCa and 11,261 healthy controls found no SNPs reaching genome-wide significance; however, three SNPs showed the same direction and a lower *p*-value in the meta-analyses than in the discovery phase. Specifically, *rs7985480* was associated with PCa risk (OR = 1.12) in linkage disequilibrium with *rs2274048*, acting by modulating binding of miRNAs to the 3’UTR of *UCHL3*, a gene involved in PCa progression, suggesting that miRNA-related SNPs are promising and useful targets for PCa risk association [78].

### 2.2. Somatic Context

The somatic genome has been an active area of research and clinical practice for several years. However, it was only in 2008 that the first report emerged using high-throughput Sanger sequencing of 20,661 protein-coding genes consolidated the four main drivers of PDAC (*KRAS*, *CDKN2A, TP53*, and *SMAD4*), which introduced the concept of core signaling pathways [79]. The identification of these four genes in the somatic genome context has shed light on molecular pathways, treatment rationale, and molecular diagnosis. Data from The Cancer Genome Atlas (TCGA) from the National Cancer Institute corroborate these data, comprising 356 samples that show mutation rates of 70.5% in *KRAS*, 61.8% in *TP53*, and 18.5% each in *SMAD4* and *CDKN2A* (National Cancer Institute, Bethesda, MD, USA). Since the progression of cancer varies due to the accumulation of mutations, one interesting area of research lies in examining the relationship between these four genes and pancreatic precursor lesions [80]. The most common precursor from which PDAC arises is pancreatic intraepithelial neoplasia (PanIN), classified by histopathology into low-grade (LG PanIN) and high-grade (HG PanIN) according to dysplasia degree [81,82]. In this context, *KRAS* and *CDKN2A* are virtually always mutated during pancreatic carcinogenesis and before invasion into pancreatic parenchyma, where the *KRAS* mutation represents the disease milestone and is correlated with the cuboidal ductal epithelium transition to columnar morphology and occurs in the early stage of PanIN [39,83]. Acquisition of alterations of *CDKN2A,* another characteristic of LG PanIN, correlates with *KRAS* and is associated with nuclear enlargement, loss of polarity, and mitotic figures [84]. The next somatic genetic alteration relies on *TP53* and *SMAD4*, which represents relatively late events that are linked with a neoplasm with lethal potential [85,86]. *TP53* alterations occur in PCa late phase and are associated with carcinoma features in situ, while *SMAD4* alterations are associated with or shortly after invasion. The alterations both typify high-grade precursors (HG PanIN) and correlate with tumor invasion [82,85].

On these genes, *KRAS* has been identified as an important genetic player since the 1980s and is the most extensively studied [87,88,89]. *KRAS* is a small GTPase acting as a molecular switch for several cellular processes and is activated by cell membrane growth factor receptors [90,91], being activated when bound to GTP and deactivated when bound to GDP. Intrinsic *KRAS* GTP–GDP cycling is regulated by guanine nucleotide exchange factors (GEFs) and by GTPase activating proteins (GAPs), which stimulate nucleotide exchange and accelerate the intrinsic GTP hydrolysis activity, respectively. Activated *KRAS* protein interacts with more than 80 downstream effector proteins and signaling pathways such as mitogen-activated protein kinase (MAPK)/MAPK kinase (MEK), phosphoinositide 3-kinase (PI3K)/AKT/mechanistic target of rapamycin (mTOR), and the rapidly accelerated fibrosarcoma (RAF)/MEK/extracellular signal-regulated kinase (ERK) pathways [92]. *KRAS* mutations are especially reliant on activating point mutations on codon 12 (exon 2), which is the initiating event in most PDAC cases (70–95%). This single-nucleotide mutation triggers the replacement of the GGT sequence (which encodes glycine) with the GAT (aspartic acid, G12D), GTT (valine, G12V), CGT (arginine, G12R), or GCT (alanine, G12A) sequences. In addition to codon 12, point mutations can also occur less frequently on codons 11, 13, 61, or 146 [93,94,95,96].

The *CDKN2A* (cyclin-dependent kinase inhibitor 2A) tumor suppressor gene is located on chromosome 9p21, whose protein controls the G1/S checkpoint, acting in cell cycle regulation by directly or indirectly targeting CDK4/6-cyclins. It encodes two unrelated proteins, sharing exons 2 and 3 but differ in exon 1: INK4 family member p16 (or p16INK4a) and p14ARF [97,98]. The first p16 arrests the cell cycle in the G1 phase by inhibiting the binding of CDK4 or CDK6 with cyclin D1, and p14ARF is related in cell cycle arrest by the p53-dependent pathway. The inactivation of *CDKN2A* mostly occurs in tandem with *KRAS* mutations and drives the malignant transformation of the pancreas [99]. Inactivation of *CDKN2A* occurs by multiple mechanisms in approximately equal proportions: homozygous deletion, mutation coupled with loss of the wild-type allele, and hypermethylation [43,100,101].

One of the most well-known and frequently mutated tumor suppressor genes across all cancers is *TP53* which encodes a protein that is known as the guradian of the genome, enrolled in modulating transcription, DNA repair, genomic stability, cell cycle control, and apoptosis [102], and it is activated by oncogenic mutations or cellular stress. As its level increases, p53 increases the transcription of downstream genes such as p21 and Bcl-2, thus, driving cell cycle arrest and repairing or eliminating damaged cells to inhibit the accumulation of oncogenic mutations [103,104]. Most alterations of *TP53* are represented by missense mutations associated with allelic loss resulting in gain of function via altered DNA binding and interactions with other transcription factors [102,105], which causes cell cycle activation, loss of apoptosis regulation, and metabolic changes [102], and phenotypes have been selected to maintain survival in association with an increasingly unstable genome [106].

Finally, the tumor suppressor gene *SMAD4* is a mediator of the canonical TGFβ signaling pathway, controlling tissue homeostasis within the pancreatic epithelium and other tissue types [107]. Although *SMAD4* is not mandatory for the activation of TGFβ signaling pathways, it is crucial for a strong signaling response. *SMAD4* shuttles between the nucleus and cytoplasm, forming a heterodimeric complex with SMAD2/SMAD3, which is phosphorylated by activated TGF-β receptors. Subsequently, this complex enters the nucleus and interacts with downstream proteins, regulating transcription of target genes [108]. Alterations in *SMAD4* occur mainly due to homozygous deletion or somatic alteration with loss of the wild-type allele [109], resulting in loss of intracellular canonical TGFβ pathway signaling, leading to increased migratory behavior, immune evasion, and autocrine activation [110].

## 3. Pancreatic Cancer Diagnosis and Prognosis by Cell-Free RNAs

### 3.1. MicroRNAs (miRNAs)

Regulatory circuits that regulate gene expression are composed of non-coding RNAs, including microRNAs (miRNAs). MicroRNAs are produced through a multistep process that culminates with the genesis of the mature miRNAs, which are single-stranded RNAs with sizes ranging between 18 and 24 nucleotides. Each mature miRNA comes from an RNA duplex with asymmetrical configuration, from which one strand will be active and the other will be degraded. Mature miRNAs integrate an RNA-induced silencing complex that can regulate the expression of target transcripts with corresponding cis-regulatory elements. MicroRNAs can act as oncogenic miRNAs (oncomiRs) or tumor suppressors in development and evolution. The identification of markers that enable the early detection of this is crucial to decrease the mortality rates of this malignancy. The stability of miRNAs in blood circulation makes them potentially useful as non-invasive biomarkers of several types of cancer [111]. In this section of the review, the current knowledge regarding the use of miRNAs as biomarkers for diagnosis and prognosis is detailed in the text and summarized in Table 1 [112].

Plasma miR-125b-3p, miR-122-5p, and miR-205-5p were shown by Marin et al. to be significantly overexpressed in the plasma exosomes of patients compared to healthy individuals. The diagnostic ability of miR-205-5p was especially important (ROC of 0.86) since it was significantly correlated with tumor progression and survival status [113].

Preoperative miR-221 concentrations in plasma are useful for detecting PCa, monitoring tumor dynamics, and predicting distant metastasis and unresectable status [114]. It has been found that circulating miR-221-3p expression levels were correlated with distant metastasis as well as with TNM stages. The diagnostic efficacy for distant metastasis of miR-221-3p was improved when compared with CA19-9 (AUC 0.689 vs. 0.587). MiR-221-3p is upregulated in PCa, promotes cell proliferation, and inhibits apoptosis [115]. The miR-221/miR-375 ratio was significantly higher in PCa patients than in healthy controls. In postoperative samples, plasma miR-221 concentrations were significantly reduced. There was also a significant correlation between high plasma miR-221 concentrations with distant metastasis and non-resectable status in PCa patients [114].

Patients were shown to highly express miR-1469-5p in both plasma and tumor tissue. Moreover, miR-1469-5p upregulation was associated with lymph node metastasis and advanced TNM stage, indicating the clinical value of this miRNA as a prognostic biomarker. Inhibition of miR-1469-5p repressed cell proliferation and invasion by targeting the metastasis suppressor NDRG1, which decreases E-cadherin expression and activates the NF-κB pathway in cells [128].

One of the members of the miR-17–92 cluster, miR-18a, has been shown to be overexpressed in plasma from patients compared to healthy individuals. Its expression was decreased in postoperative samples. Accordingly, miR-18a has been shown to be overexpressed in tissues and cell lines. Plasma miR-18a had an AUC of 0.9369, demonstrating its potential to distinguish patients from healthy individuals [116].

One of the most promising and explored miRNAs in PDAC is miR-21 which is also overexpressed in different types of cancer. The oncogenic role of miR-21 is demonstrated by the targeting of diverse tumor suppressor genes, such as phosphatase and tensin homolog (PTEN), tropomyosin 1 (TM1), programmed cell death 4 (PDCD4), and tissue inhibitor of metalloproteinase 3 (TIMP3). In PDAC, miR-21-5p is upregulated compared to normal controls, even in microscopic precancerous pancreatic lesions such as non-invasive pancreatic intraepithelial neoplasia and in benign pancreatic lesions. Plasma levels of miR-21-5p are significantly higher in PDAC and in intraductal papillary mucinous neoplasm (IPMN) compared to normal controls, as reviewed by [129]. An analysis of serum exosome miRNA content revealed the upregulation of miR-191, miR-21, and miR-451a in patients compared to the controls. The area under the curve and the diagnostic accuracy of exosomal miRs were 5–20% greater than those of three serum bulky circulating miRs. Among the three, exosomal miR-21 showed the largest AUC and highest diagnostic accuracy (AUC 0.826) [117]. Kawamura et al. described an increased expression of miR-4525, miR-451a, and miR-21 in the plasma of stage II PDAC patients who experienced recurrence after surgery, compared to the recurrence-free patients and healthy controls. The increased expression of the same three miRNAs also occurred in portal vein blood (PVB), but with increased sensitivity, specificity, and accuracy compared to peripheral blood. Notably, the three miRNAs were shown to be independent prognostic factors for overall and disease-free survival [130]. Pu et al. (2020) described a greater expression of plasma exosomal miR-21 and miR-10b in patients with PCa compared to healthy individuals. The diagnostic value of exosomal miR-21 was improved when combined with exosomal miR-10b (*p* < 0.0001, AUC 0.791). Exosomal miR-21 was capable of distinguishing patients with early-stage PCa from controls and advanced-stage PCa [118]. Additionally, Alemar et al. (2016) described increased serum levels of miR-21 and miR-34a in PDAC compared to healthy controls. The AUCs for miR-21 and miR-34a were 0.889 and 0.865, respectively. Using the optimal cutoff point, the sensitivity and specificity of miR-21 were 82.6% and 77.8% and those of miR-34a were 91.3% and 77.8%, respectively. A combined ROC curve for miR-21 and miR-34a resulted in subtle improvement (AUC 0.894). Therefore, the combination of circulating miR-21 and miR-34a clearly discriminated patients with PDAC from healthy controls with sufficient sensitivity and specificity [119].

Shao et al. (2021) reported that circulating levels of miR-483-3p were higher in the serum and serum exosomes of PDAC patients compared to healthy individuals. Serum miR-483-3p levels could distinguish PDAC patients from healthy individuals with a ROC curve area (AUC) of 0.81 (74.6% and 77.3% sensitivity and specificity, respectively). Moreover, serum miR-483-3p levels were able to detect early-stage (≤2 cm) PDAC with an AUC of 0.83 and sensitivity and specificity of 85.7% and 72.7%, respectively. Interestingly, the diagnostic value of the serum miR-483-3p level was greater than that of exosomal miR-483-3p levels (AUC of 0.69). In addition, higher serum exosome miR-483-3p levels predicted worse survival and were an independent prognostic factor for PDAC. The expression levels of miR-483-3p were higher in PDAC and pancreatic intraepithelial neoplasia tissues (PanIN) and negatively correlated with *SMAD4* expression, suggesting that miR-483-3p may exert oncogenic functions in the early pathogenesis of PDAC. Analysis of formalin-fixed, paraffin-embedded (FFPE) tissue sections revealed miR-483-3p expression in 64.1% of PanIN-1, 84.5% of PanIN-2, and 96.6% of PanIN-3 lesions, while it reached 100% expression in PDAC lesions. Importantly, miR-483-3p was not expressed in normal pancreatic ducts. A mutually exclusive expression of miR-483-3p and *SMAD4* was described in PDAC tissues, the adjacent PanIN lesions, and normal pancreatic ducts, reinforcing previous data indicating that miR-483-3p targets *SMAD4*. In addition, miR-483-3p and *SMAD4* protein expression were negatively correlated in both PanIN and PDAC lesions (γ = −0.770, *p* < 0.0001), which led to the conclusion that miR-483-3p inhibits *SMAD4* during the development of PDAC [120].

The potential use of miR-1246 as a biomarker for diagnosis has been reported for hepatocellular carcinoma (HCC), ovarian cancer, and esophageal cancer. Ishige et al. reported significantly higher miR-1246 expression in the serum and urine of pancreatic cancer patients relative to healthy controls. The AUC for serum miR-1246 was 0.87 (sensitivity 92.3% and specificity 73.3%) to distinguish between PCa patients and healthy individuals. The expression of the tumor suppressor CADM1 is downregulated by the miR-1246 network in HCC cell lines, thereby enhancing cell migration and invasion. Importantly, miR-1246 expression was associated with cancer cell stemness and chemoresistance through its targeting of the tumor suppressor cyclin G2 [121].

Yu et al. (2020) reported increased levels of miR-25 in the serum of PCa patients relative to non-cancer controls. The combination of CA19-9 and miR-25 had a higher diagnostic sensitivity in the early stages than both CA19-9 alone and the combination of CA19-9 and CA125, which is widely used in the diagnosis and prognosis of PCa. Thus, serum miR-25 may be a promising predictive marker for PCa, with particular efficacy for early PCa diagnosis when combined with CA19-9. The AUC for serum miR-25 yielded a value of 0.939 (95% CI 0.903–0.975), which indicated that it can be used as a diagnostic biomarker, differentiating PCa patients from healthy individuals. Likewise, higher miR-25-3p levels were reported in PDAC as compared to non-tumor tissues. In vivo and in vitro models of PDAC showed that the overexpression of miR-25-3p promoted cell proliferation and metastasis. PH domain leucine-rich repeat protein phosphatase 2 (PHLPP2) was shown to be suppressed by mature miR-25, which led to the activation of oncogenic AKT-p70S6K signaling in PCa cells [122].

Miyamae and colleagues (2015) selected miR-744 for further study after screening the expression levels of 1719 miRNAs in the plasma of PCa patients and healthy volunteers using a 3D-Gene microRNA array-based approach. The higher expression of miR-744 in the plasma of PCa patients compared to healthy volunteers was validated in a small-scale analysis, two independent cohort analyses, and a large-scale analysis (AUC 0.831). Moreover, expression of miR-744 was significantly reduced in plasma after pancreas tumor resection. There was a positive correlation between high expression of plasma miR-744 and lymph node metastasis and recurrences, which makes it a poor independent prognostic factor of PCa patients after pancreatectomy. Furthermore, PCa patients with high levels of plasma miR-744 tended to have progressive invasiveness from IPMN carcinoma to PDAC. No highly sensitive predictive biomarker of branch duct-type IPMN malignancy is currently known, highlighting the importance of validating plasma miR-744 as a marker of invasiveness in PCa before surgery. Therefore, miR-744 could contribute to screening PCa and monitoring tumor dynamics [123].

Zou et al. (2019) described the upregulation of six miRNAs (let-7b-5p, miR-192-5p, miR-19a-3p, miR-19b-3p, miR-223-3p, and miR-25-3p) in the serum of PCa patients relative to healthy individuals, after an early screening and subsequent validation phases. The panel was able to discriminate PCa patients from healthy controls (AUC of 0.978, sensitivity = 93.3%, specificity = 96.0%). Among the six miRNAs, higher serum miR-19a-3p levels were an independent predictor of worse OS, alongside other influencing factors, including vascular/nerve infiltration and positive lymph nodes. Tissue and serum-derived exosomes showed significantly higher levels of miR-192-5p, miR-19a-3p, and miR-19b-3p in PCa patients than in healthy controls [124]. Flammang et al. (2020) described how serum exosome-derived miR-192-5p was able to distinguish healthy individuals from PDAC patients (AUC of 0.83, *p* = 0.0004) and from patients with CP (AUC of 0.80; *p* = 0.0164), but not PDAC from CP patients (AUC of 0.54, *p* = 0.7206). Serum-free miR-192-5p could not differentiate between any of these three groups of patients. Interestingly, the researchers found that miR-192-5p was significantly downregulated in PDAC tumors compared to healthy tissue [125].

Khan et al. [126] performed NGS and identified 219 differentially expressed miRNAs in pancreatic tissue specimens from autopsy cases of patients with PDAC and chronic pancreatitis (CP), as well as from normal pancreatic tissue. They selected the eight most differentially expressed miRNAs between PDAC and CP for further validation using qRT-PCR: the four most upregulated (miR-215-5p, miR-22-5p, miR-192-5p, and miR-181a-2-3p) and the four most downregulated miRNAs (miR-30b-5p, miR-216b-5p, miR-320b, and miR-214-5p). On the one hand, the expressions of three miRNAs (miR-215-5p, miR-122-5p, and miR-192-5p) were confirmed in serum samples, which showed an upregulation in PDAC patients compared to CP patients and healthy controls. On the other hand, miR-30b-5p and miR-320b were identified as downregulated miRNAs in PDAC versus CP. Five of the eight miRNAs were validated in serum samples as potentially strong biomarkers for the early detection of PDAC. The expressions of miR-215-5p, miR-122-5p, and miR-192-5p in PDAC tissues compared to both CP tissue and normal pancreatic tissue suggest their role in tumor progression, despite contradictory reports about the function of these miRNAs in various cancers [125].

Ye et al. (2020) performed an miRNA array-based analysis of 372 miRNAs and identified 28 miRNAs that were differentially expressed in PDAC compared to healthy controls. By comparing gemcitabine-resistant and -sensitive groups, 24 miRNAs showed significant changes. However, the authors decided to focus on miR-7 because its expression level decreased not only in PDAC compared to healthy controls but also in the gemcitabine-resistance group compared to the gemcitabine-sensitive group. Lower miR-7 expression was significantly correlated with advanced tumor stage anda worse prognosis. Absent or lower miR-7 expression was associated with poor prognosis, poor tumor differentiation, advanced TNM stage, and distant metastasis [131].

Duell et al. [132] published a prospective cohort study with samples collected years before the diagnosis of PDAC within the European Prospective Investigation into Cancer and Nutrition cohort (EPIC). The final cohort consisted of 225 PDAC cases and 225 matched normal controls. Based on adjusted logistic regression models, levels of four miRs (-10a, -10b, -21-5p, and -30c) at follow-up time between blood collection and diagnosis of ≤five years and ≤two years were statistically significantly associated with a risk of developing PDAC [129]. Xue et al. [133] reviewed 29 studies assessing the potential of circulating miRNAs as non-invasive diagnostic biomarkers. From a total of 68 evaluated miRNAs, the authors reported miRNAs individually or in panels as potential diagnostic biomarkers. Among those, the most frequently found in the studies were miR-20a-5p, miR-21-5p, miR-22-3p, miR-22b-3p, and miR-885-5p [133].

Gablo et al. (2020) performed RNA sequencing of preoperative plasma from 112 patients with PDAC divided into cohorts of discovery (*n* = 48) and validation (*n* = 64) and further subdivided into poor prognosis (OS shorter than 16 months) and good prognosis (OS longer than 20 months) for the discovery phase. The validation confirmed that higher preoperative levels of miR-99a-5p and miR-365a-3p were associated with better survival of PDAC patients who underwent curative surgery. MiR-99a-5p has been described as both an oncogene and a tumor suppressor. A low expression of miR-356a-3p in pancreatic tissue has been linked to increased PDAC progression. Increased expression of miR-365a-3p correlates negatively with c-Rel expression and inhibits NF-κB activity, which reduces viability and induces apoptosis of PDAC cells [134].

Lai and colleagues reported that plasma exosome levels of miR-10b, -21, -30c, -181a, and -let7a had 100% sensitivity and specificity with respect to their accuracy in distinguishing PDAC patients from controls. Similar trends have been observed for these miRNAs in plasma. The accuracy in distinguishing PDAC from healthy controls was 100% (sensitivity and sensibility) for miR-10b and miR-30c. Exosomal miR-106b had an AUC of 0.85, while plasma miR-106b had an AUC of 0.98; thus, plasma was shown to be more sensitive for differentiating PDAC from normal samples [127].

Serum miR-373-3p was highly expressed in PDAC patients with progressive disease before the start of FOLFIRINOX and serum miR-194-5p expression was decreased after one cycle of FOLFIRINOX, compared to healthy controls; both miRNAs were significant predictors of early tumor progression during FOLFIRINOX [135].

### 3.2. Long Non-Coding RNAs (lncRNAs)

Recently, lncRNAs have been widely studied, and their considerable potential for diagnosing several types of cancers has been demonstrated [136,137,138]. LncRNAs are composed of at least 200 nucleotides, are easily detectable in human body fluids, and their binding and expression are highly specific. Several recent studies have shown that the differential expression of lncRNAs is related to PCa [139,140,141]. Throughout this topic, cell-free lncRNAs with potential application in the diagnosis and prognosis of this malignant disease will be discussed and summarized in Table 2.

Hox transcript antisense RNA (HOTAIR) is a long non-coding RNA involved in the pathogenesis of numerous types of cancer. It has been found to be upregulated in PCa tissue compared to adjacent tissues in serum samples of PCa patients compared with healthy controls [142]. Furthermore, HOTAIR expression increases with tumor progression, making it a valuable biomarker candidate that may be used for diagnosis and prognosis of pancreatic adenocarcinoma [142]. It has been found that HOTAIR upregulation promotes hexokinase 2 (HK2) expression, a protein that plays a pivotal role in energetic cancer metabolism. Thus, HOTAIR may promote cancer cell energy metabolism by upregulating HK2 expression [142].

Liu et al. (2021) selected eleven lncRNAs related to PCa from the Cancer Genome Atlas (TCGA) database and evaluated their expression in 30 non-cancer and 15 PCa patients. From these eleven lncRNAs, three were selected for further validation in a larger cohort—ABHD11-AS1, LINC00176, and SNHG11—since they were upregulated in PCa patients compared to healthy individuals. Plasma ABHD11-AS1 was confirmed to be an excellent biomarker with the best diagnostic performance for the early detection of PCa [143]. Likewise, a later study observed that ABHD11-AS1 was negatively correlated with the survival rates of patients with PCa and that ABHD11-AS1 silencing significantly inhibited the proliferation and induced the apoptosis of PCa cells [146].

LINC01111 has been found to be downregulated in tissue and plasma samples of patients with PCa; in addition, it plays a role in tumor suppression and has been positively correlated with the survival of PCa patients. It acts to sequester miR-3924, leading to the upregulation of dual specificity phosphatase 1 (DUSP1), which causes blockage of SAPK phosphorylation and inactivation of the SAPK/JNK signaling pathway, inhibiting PCa aggressiveness [147].

Another lncRNA present in increased levels in serum samples of PCa patients compared with those of healthy individuals is UFC1. An ROC curve analysis revealed an AUC value of 0.810, highlighting this lncRNA as a promising PCa biomarker. UFC1 expression was related to lymph node metastasis, distant metastasis, and clinical stage [144].

Kumar et al. described a higher expression of the lncRNAs MALAT1 and CRNDE in serum exosomes of PDAC or IPMN patients compared to healthy individuals [148]. MALAT1 acts as a transcriptional and epigenetic regulator [149], while CRNDE modulates cell proliferation and angiogenesis via the miR-451a/CDKN2D axis in PCa, representing a possible therapeutic target for PCa treatment [150].

A large study profiling lncRNA expression using RNAseq in plasma extracellular vesicles of 284 PCa patients and 117 healthy controls was performed by [122]. They found that FGA, KRT 19, HIST1H2BK, ITIH2, MARCH2, CLDN1, MAL2, and TIMP1 were good biomarkers for PDAC detection, showing high accuracy with an AUC of 0.960, sensitivity of 93.39%, and specificity of 85.07%. CA19-9 is the biomarker currently used for PDAC detection [151], and the set of lncRNA biomarkers described by the authors was able to distinguish CA19-9 negative PDAC from controls and PDAC from chronic pancreatitis. This was the first characterization of lncRNAs in extracellular vesicles, and the signature found was capable of diagnosing PCa and improving prognostics [145].

SNHG15 is a lncRNA involved in tumor malignancy behavior and has been indicated to be a potential biomarker for pancreatic adenocarcinoma diagnosis and prognosis [152]. Its levels were found to be upregulated in 171 PDAC serum samples compared with 59 healthy patients. SNHG15 overexpression was associated with tumor size, tumor node metastasis stage, and lymph node metastasis in patients with PC. This lncRNA inhibits P15 and Kruppel-like factor 2 (KLF2) expression to promote PCa proliferation through EZH2-mediated H3K27me3 [153].

Few studies have demonstrated the potential of circulating lncRNAs as biomarkers for PCa, although their usefulness in the diagnosis and prognosis of the disease is clear. The search for these biomarkers should continue, and finding some with good sensitivity and specificity for a particular type of cancer is certainly a breakthrough in the discovery and treatment of the disease in view of its relatively non-invasive nature.

### 3.3. Circular RNAs (circRNAs)

Circular RNAs (circRNAs) are a type of single-stranded, closed-loop RNA structure that originate from pre-mRNAs during transcription and were initially proposed to be by-products of splicing or splicing errors. Recently, the remarkable role of circRNAs has been described in cancer cell proliferation, differentiation, apoptosis, and metastasis. The dysregulation of circRNAs has been described in different malignancies, indicating their potential utility in cancer diagnosis and prognosis. In this section, the current knowledge regarding the use of circRNAs as biomarkers of PCa is detailed in the text and summarized in Table 3.

In PCa, by using using RT-qPCR, Circ_001569 has been found to be upregulated in the plasma of PCa patients compared with healthy controls. High Circ_001569 levels were positively correlated with lymphatic metastasis and clinical stage and had a poor prognosis. The sensitivity and specificity of this circRNA as a biomarker of PCa were 62.76% and 74.29%, respectively. In addition, in vitro silencing of Circ_001569 decreased cell proliferation, migration, and invasion [154].

The diagnostic value of circular RNA circ-LDLRAD3 as a biomarker in the diagnosis of PCa was analyzed with RT-qPCR using 31 plasma samples of PCa patients and 31 plasma samples from healthy volunteers. Additionally, 30 paired PCa tissues and adjacent non-tumor tissues were analyzed. Circ-LDLRAD3 was upregulated in PCa tissues (*p* < 0.01) and plasma samples from patients with PCa (*p* < 0.01). High levels of this circRNA were correlated with venous invasion, lymphatic invasion, and metastasis. The area under the ROC curve was 0.67 and increased to 0.87 when combined with CA19-9. The sensibility and specificity reached were 0.57% and 0.70%, respectively. This data suggest that circ-LDLRAD3 may be helpful in PCa diagnosis [155]. The circRNA hsa_circ_0013587 was also found to be upregulated in the plasma of PCa patients (*n* = 60) compared with plasma from healthy controls (*n* = 60). In addition, hsa_circ_0013587 was upregulated in PCa patients with later clinical stages III–IV compared to those detected in early clinical stages I–II [156].

Liu et al. (2021) analyzed plasma samples from 38 patients with PDAC and 38 healthy volunteers. Using RT-qPCR, they found that circRNA_00068 was significantly upregulated in PDAC plasma compared with healthy individuals. In addition, to evaluate the function of this circRNA, they overexpressed or silenced its expression in ANC-1 and AsPC-1 cells and found that elevated levels of circRNA_00068 enhanced proliferation, migration, invasion, and angiogenesis of PDAC cells [160].

There are a variety of circular RNAs present in exosomes, but their function in cancer development is still unclear [162]. Li et al. examined 85 tissue samples and exosome plasma from patients with PDAC and found that circ-IARS was upregulated in both types of samples from patients with metastatic disease. Furthermore, they found that circ-IARS enters human microvascular vein endothelial cells through exosomes and acts as a sponge to absorb miR-122. This action leads to increased monolayer permeability due to increased activity of RhoA, expression of F-actin, and reduced expression of Z0-1, promoting tumor metastasis. These findings provide arguments for the investigation of circ-IARS in promoting tumor invasion and metastasis and provide evidence for this circRNA to be a possible diagnosis and prognosis biomarker [157].

Other circRNAs found to be upregulated in the plasma exosomes from PCa patients are hsa_circ_0006220 and hsa_circ_0001666. The AUC values were 0.7817 for hsa_circ_0006220 and 0.8062 for hsa_circ_0001666, increasing to 0.884 when both circRNAs were combined. This analysis was performed with plasma exosomes from 62 patients with PCa and 62 healthy volunteers. Moreover, hsa_circ_0001666 expression was linked with tumor size and CA-19-9 levels, and the expression of hsa_circ_0006220 was associated with CA19-9 levels and lymph node metastasis [158].

Li et al. (2018) analyzed circ-PDE8A expression in plasma exosomes of PDAC patients. They found that exosomal circ-PDE8A was associated with progression and prognosis in PDAC patients playing a pivotal role in tumor invasion [159]. Another study found that circPDK1 was highly abundant in PCa tumor tissues and serum exosomes and was associated with poor survival. CircPDK1 is transcriptionally activated by HIF1α and sponges miR-628-3p to activate the BPTF/c-myc axis [161]. BPTF is necessary for c-MYC-driven proliferation, and c-MYC is an oncogene deregulated in most human tumors [163].

## 4. Circulating Cell-Free Tumoral DNA (ctDNA)

One of the most interesting molecules for a liquid biopsy is circulating cell-free DNA (ccfDNA), which is comprised of double-strand DNA fragments (70–200 bp) circulating in almost all biological fluids (blood plasma, serum, pancreatic juice, bile, and urine), which can be released from circulating tumor cells (CTCs), circulating tumor DNA (ctDNA), and normal cells [164,165]. In liquid biopsy, RNA from extracellular vesicles (EVs) and tumor-educated platelets (TEPs) are also targets, but the focus here is on DNA.

Circulating tumor cells (CTCs) were first described in an autopsy by Ashworth [166] who found cells in blood that were identical to the tumor cells in skin lesions, suggesting that the cells must pass through the circulatory system for tumors to metastasize. Subsequent studies have corroborated this characterization [167,168]. Leon et al. (1977) showed a correlation between patients with malignancy and the absolute concentration of ccfDNA in their serum. They demonstrated that elevated concentrations of ccfDNA from patients with several types of tumors decreased after radiation therapy [169].

Nowadays, it is accepted that CTCs indicate tumor progression and an increased risk of metastasis, and that early CTC detection can identify patients with a high risk of metastasis and help in customizing adjuvant therapies for patients in advanced stages of PDAC [164]. CTCs have been found in some types of cancer, such as breast, prostate, colorectal, ovarian, and lung; however, it is difficult to obtain CTCs. This obstacle has been be circumvented using a cell-free fraction of blood to extract DNA (ccfDNA), which has also facilitated studies about tumor mutations [170]. Nevertheless, only the presence and concentration of ccfDNA are insufficient to guide clinical decisions since many other healthy conditions can alter levels of ccfDNA [164], such as exercise, sepsis, inflammatory conditions, and tissue injury [171].

One type of ccfDNA molecules is ctDNA, which is released from tumor cells. These molecules are of interest since they represent a promising approach for accessing tumor DNA, and some of these molecules might be used as tumor-specific biomarkers of tumor DNA [164]. Circulating tumor DNA allows easy and early access to information on tumor evolution and treatment response in a non-invasive manner, enabling improved therapy decisions and improving patients’ quality of life [164,172]. Therefore, this type of molecule is an important target in liquid biopsy studies. Moreover, ctDNA comprises apoptotic or necrotic parts of the primary tumor, as well as metastasis and CTCs. It can be distinguished from normal ccfDNA by the presence of cancer-specific mutations [173,174], so the fraction of ctDNA in ccfDNA is extremely small, especially in the early stages and after positive therapies. Therefore, detecting ctDNA is a major challenge when attempting to use it in tumor diagnosis and prognosis, which is being overcome with considerable effort [164,173]. Analyses using highly sensitive and specific methods are important for distinguishing ctDNA from normal ccfDNA, such as specific somatic mutations, structural variations, and epigenetic patterns [175].

It is suggested that ctDNA detection can be used as a non-invasive blood biomarker with different purposes: early detection, prognosis estimation, treatment selection, tumor dynamics monitoring, minimal residual disease (MRD), and tumor recurrence [176]. Some studies have reported ctDNA detection in early-stage PDAC so that it can be used in diagnosis [177], although multiple points should be measured to access all the information in ctDNA. Plasma nucleases can degrade ctDNA, and therefore, the time between the blood collection and plasma isolation by centrifugation is very critical: This process should ideally be performed within less than 1 h [173]. However, other studies have shown that ctDNA is stable for more time, depending on the kind of tube in which the blood was collected and the presence of stabilizers [178].

Circulating tumor DNA can be detected by targeting known tumor-specific mutations (such as *KRAS* and others) or looking for de novo genetic alterations by investigating multiple genes simultaneously (NGS approaches). For the first, digital PCR (dPCR) or droplet digital PCR (ddPCR) are promising approaches due to their precision, sensitivity, and specificity. To identify new alterations, NGS is the approach used to enable ctDNA study without knowing the genotype of the tumor. However, NGS has lower sensitivity than dPCR or ddPCR [164].

There are some ctDNA tests that can be used in many applications, including early screening, detection of known mutations to predict treatment response, MRD, and monitoring therapy results. One of the more acceptable tests is CancerSeek [177,179], validated in 1005 patients of eight types of stage I-III cancer (breast, colorectal, gastric, liver, lung, esophageal, ovarian, and PCa). This test uses common circulating proteins and DNA mutations from these types of cancer, containing a 61-amplicon (about 33 pb) panel within one of 16 target genes. About 70% of these patients were positive for cancer using the CancerSeek protocol, with a sensitivity of >69% for PCa and specificity of 99% for all types of cancer involved in the study. Another positive point for CancerSeek is that the use of a supervised machine-learning algorithm for the multi-analyte data enabled the correct identification of the organ of origin in 63% of the positive patients. However, in early-stage detection, which is frequently asymptomatic, the sensitivity was lower: 43% for stage I. Even so, this test is innovative, and further studies may be a starting point for a non-invasive blood-based diagnosis for some kinds of solid tumors.

Kinetics is an important consideration in the ctDNA context since there are quantitative changes in quantities of these molecules over time and because many factors can contribute to these changes, such as tumor biology, host physiology, and treatment [171,180]. In the initial stages of PDAC there are low quantities of ctDNA, so using it in diagnosis requires special attention. A recent meta-analysis of some studies that utilized ctDNA as a biomarker for diagnostic PDAC by liquid biopsy showed a sensitivity of 0.80 (95% CI 0.77–0.82), specificity of 0.89 (95% CI 0.87–0.91), and an AUC of 0.936. With these results, they concluded that liquid biopsy can be used to detect PDAC [181]. This same meta-analysis was performed with exosomes and showed greater parameters: sensitivity, specificity, and AUC of 0.93 (95% CI 0.90–0.95), 0.92 (95% CI 0.880.95), and 0.9819, respectively, showing that exosomes have a strong diagnostic value.

In the somatic mutation context, specific mutations may be available, such as *KRAS, TP53,* and *CDK2NA*. The parameters of interest are the detection of mutations and/or quantitative analysis of the mutant allelic fraction. These parameters are relevant for monitoring treatment response, disease burden, and outcome [165]. *KRAS* mutations are the most common target used for PDAC detection. Many studies have reviewed the available data in the literature and shown this [165,182,183,184,185]. *KRAS* mutations are rarely found in clones during age-associated clonal hematopoiesis, such as in DNMT3A, TET2, JAK2, ASXL1, *TP53*, GNAS, PPM1D, BCORL1, and SF3B1. For this reason, it is a good candidate for early-detection cancer screening. However, since the genes mentioned above are a potential source of false–positive results, their use as biomarkers is not recommended [186,187,188]. A PCa meta-analysis (14 studies with 369 patients) examined *KRAS* mutations and showed that the sensitivity and specificity of liquid biopsy to diagnosis compared with molecular tissue analysis specimens were 65% and 91%, respectively [189]. This same study examined the sensitivity and specificity in the diagnostic accuracy parameters for liquid biopsy compared to tissue specimens and showed 70% and 86%, respectively. The SROC curves indicated that, compared to tissue specimens, liquid biopsy demonstrated a high accuracy in determining the mutational potential of PDAC (AUC of 0.880 for all studies and 0.882 for those regarding *KRAS* only). However, although *KRAS* mutation is a great indication of PDAC, it can be found in other types of cancer. The specificity in this study reflects the correlation between the primary tumor tissue and the liquid biopsy and not the overall specificity of *KRAS* mutational status for PDAC diagnosis. In a PDAC study applied in metastatic patients, after two weeks of antineoplastic treatment, the levels of *KRAS* ctDNA decreased by 57.9%, with 100% specificity and 91.67% sensitivity (AUC = 0.918), suggesting that *KRAS* ctDNA can be used as a biomarker. However, the pretherapeutic ctDNA detection was associated with worse OS in chemotherapy patients, independent of the treatment line evaluated [180].

There are many studies that have shown *KRAS*, *TP53*, *SMAD4*, and *CDKN2A* detection by dPCR or ddPCR in localized PDAC, suggesting that the latter may be useful in prognosis [190,191,192,193,194,195,196,197,198,199]. All these studies use a G12/G13 mutation panel as a target for *KRAS* mutations, which includes the regions mutated in PDAC by NGS or dPCR. As reviewed by [200], exo*KRAS* have demonstrated greater sensitivity and specificity in predicting disease progression compared to ctDNA *KRAS* [201,202]. Allenson et al. [203] found that for predicting PDAC status, the sensitivity and specificity were 75.4% and 92.6%, respectively, and positive *KRAS* mutation from exoDNA was significantly associated with early-stage PDAC. The authors focused on exosomes, which are synthetized by specific pathways and not released by apoptosis or necrosis; therefore, exoDNA was associated with early stages of cancer, while ccfDNA was associated with later stages of the disease [202].

In 2017, Cohen et al. suggested that the use of *KRAS* mutation and CA19-9 detection associated with four other protein biomarkers could improve the sensitivity of a noninvasive blood test for PDAC detection. In this study, this combination detected 64% of resectable cancer (95% CI 57–70%), decreasing to 49% when only CA19-9 was used (95% CI 43–56%). Another important result of this study was its high specificity (99.5%, 95% CI 97–100%), which is important for a diagnostic test. In 2020, Macgregor-Das et al. found parameters for diagnostic performance of *KRAS* mutation combined with CA19-9 in plasma samples. The combined sensitivity of both biomarkers was 66.7% (95% CI 51.1–80%).

Epigenetic markers in blood ccfDNA are a promising biomarker for DNA [165], since methylation patterns are more studied than histone modifications. Some recent studies have shown that DNA methylation signatures were consistent between ccfDNA and the genomic DNA from its tissue origins, also in cancer models [204,205]. Relative to somatic mutations, there is high heterogeneity found in the tumor, but because of the similar epigenetic profiles, they can help identify tumor origin with liquid biopsy and in metastatic settings [204]. Many studies have shown that promoter methylations can be used as biomarkers for early-stage PDAC detection and can be used to differentiate PDAC from pancreatitis with sensitivity and specificity of 91.2% and 90.8%, respectively [206,207]. The 5-methylcytosine (5 mC) modification is the most abundant form of DNA methylation and is involved in gene regulation expression. It is being studied as an epigenetic ccfDNA biomarker and once this is a tissue-specific modification it can be used to monitor the tumor burden [208].

## 5. Conclusions

Despite all the difficulties and questions around using ctDNA as a biomarker, some clinical trials (clinicatrials.gov accessed on 27 February 2023; [209]) are being developed or have finished that are looking for ctDNA liquid biopsy biomarkers for PCa prognosis or prediction. There are many questions regarding the use of CTCs and ctDNA as biomarkers: What influences the fluctuation levels of CTCs or ctDNA in blood? How can we differentiate DNAs released from resistance cells or treatment response? and Are the extraction and detection methods adequate to give sensitivity to the results? Some studies have shown the pros and cons of their use in liquid biopsies. The major benefit is their high specificity for specific mutations, mostly in the advanced stages of cancer; the major drawback is their low sensitivity in early-stage cancers. However, their use is important for prognoses in many metastatic settings, prediction of relapse after treatment as well as therapy resistance, drug screening and functional analysis using live CTCs, tumor evolution, and early detection of druggable mutations [164]. Characterizing CTCs is important for developing new treatments, as well as for deciphering the metastatic process, but the EMT process alters the membrane protein pattern of the CTCs, influencing their detection. One of the important steps in cancer development is the epithelial–mesenchymal transition (EMT) since the cells lose their phenotype, acquiring a mesenchymal phenotype that is more invasive and proliferative and eventually increases cancer cell stemness [173].

Therefore, this review shows the relevance of pancreatic cancer worldwide, and provides background information on diagnostic complexity and its importance related to progression and mortality rate. The germline and somatic contexts of pancreatic cancer were both described, which are linked with circulating cell-free nucleic acids as promising and novel diagnostic and prognostic tools. A pancreatic screening protocol to identify these molecules can provide a tool for earlier diagnosis compared to that of conventional diagnostic methods, allowing more adapted and assertive treatments that may have an impact by reducing morbidity and mortality rates.

## Figures and Tables

**Table 1 biomedicines-11-01069-t001:** Summary of promising microRNAs for diagnosis of pancreatic cancer.

miRNA	Comparison vs. Healthy Control	Sample	Regulation	AUC	Ref.
miR-205-5p	PCa	plasma exosome	up	0.860	[113]
miR-221	PCa	plasma	up	0.743	[114]
miR-221-3p	PCa	plasma	up	0.689	[115]
miR-375	PCa	plasma	down	0.573	[114]
miR-18a	PCa	plasma	up	0.9369	[116]
miR-21	PCa	serum exosome	up	0.826	[117]
miR-21	PCa	serum	up	0.653	
mir-21	PCa	plasma exosome	up	0.717	[118]
miR-21	PDAC	serum	up	0.889	[119]
miR-21	PDAC	plasma exosome	up	1.00	
miR-21	PDAC	plasma	up	0.95	
miR-34a	PDAC	serum	up	0.865	
miR-483-3p	PDAC	serum	up	0.830	[120]
miR-191	PCa	serum exosome	up	0.788	[117]
miR-191	PCa	serum	up	0.604	
miR-415a	PCa	serum exosome	up	0.759	
miR-415a	PCa	serum	up	0.518	
miR-1246	PCa	serum	up	0.870	[121]
miR-25	PCa	serum	up	0.939	[122]
miR-744	PCa	plasma	up	0.831	[123]
let-7b-5p	PCa	serum	up	0.703	[124]
miR-192-5p	PCa	serum	up	0.684	
miR-192-5p	PDAC	serum exosome	up	0.830	[125]
miR-192-5p	PCa	serum exosome	up	0.800	
miR-19a-3p	PCa	serum	up	0.771	[124]
miR-19b-3p	PCa	serum	up	0.788	
miR-223-3p	PCa	serum	up	0.901	
miR-25-3p	PCa	serum	up	0.726	
miR-122-5p	PDAC	serum	up	0.988	[126]
miR-320b	PDAC	serum	up	0.922	
miR215-5p	PDAC	serum	up	0.832	
miR-10b	PDAC	plasma exosome or plasma	up	1.00	[127]
miR-30c	PDAC	plasma exosome or plasma	up	1.00	
miR-181a	PDAC	plasma exosome	up	1.00	
miR-181a	PDAC	plasma	up	0.97	
miR-let7a	PDAC	plasma exosome	up	1.00	
miR-let7a	PDAC	plasma	up	0.99	
miR-106b	PDAC	plasma exosome	up	0.86	
miR-106b	PDAC	plasma	up	0.98	

PCa, pancreatic cancer; PDAC, pancreatic ductal adenocarcinoma; AUC, area under the curve.

**Table 2 biomedicines-11-01069-t002:** Summary of promising long non-coding RNAs for diagnosis of PCa.

lncRNA	Comparison vs. Healthy Control	Sample	Regulation	AUC	Ref.
HOTAIR	PDAC	serum	up	0.933	[142]
ABHD11-AS1	PCa	plasma	up	0.887	[143]
LINC00176	PCa	plasma	up	0.707	
SNHG11	PCa	plasma	up	0.790	
UFC1	PCa	serum	up	0.810	[144]
Panel composed by FGA, KRT19, HIST1H2BK, ITIH2, MARCH2, CLDN1, MAL2 and TIMP1	PDAC	Plasma exosome	up	0.960	[145]

PCa, pancreatic cancer; PDAC, pancreatic ductal adenocarcinoma; AUC, area under the curve.

**Table 3 biomedicines-11-01069-t003:** Summary of promising circular RNAs for diagnosis of pancreatic cancer.

circRNA	Comparison vs. Healthy Control	Sample	Regulation	AUC	Ref.
Circ_001569	PCa	plasma	up	0.716	[154]
circ-LDLRAD3	PCa	plasma	up	0.670	[155]
hsa_circ_0013587	PCa	plasma	up	0.801	[156]
circ-IARS	PDAC	plasma exosome	up	-	[157]
hsa_circ_0006220	PCa	plasma exosome	up	0.781	[158]
hsa_circ_0001666	PCa	plasma exosome	up	0.806	
Circ-PDE8A	PDAC	plasma exosome	up	-	[159]
circRNA_00068	PDAC	plasma plasma	up	-	[160]
CircPDK1	PDAC	serum exosome	up	-	[161]

PCa, pancreatic cancer; PDAC, pancreatic ductal adenocarcinoma; AUC, area under the curve.

## Data Availability

No new data were created or analyzed in this study. Data sharing is not applicable to this article.

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
