# Peer review of "Circulating Cell-Free Nucleic Acids as Biomarkers for Diagnosis and Prognosis of Pancreatic Cancer"

_biomedicines, 2023, doi:10.3390/biomedicines11041069_

Round 1
Reviewer 1 Report
In this manuscript, Marin et al. reviewed progress in prognosis of pancreatic cancer by biomolecules from liquid biopsy. The authors put emphasis on circulating cells and nucleic acids in human blood. The potential of using these circulating cells or nucleic acids as biomarkers for pancreatic cancer diagnosis and therapy management have also been carefully discussed. The results summarized in this manuscript covered detailed original studies of genetic factors and circulating cells and biomolecules in early detection and management of pancreatic cancers.
This literature review focused on key diagnostic questions in pancreatic cancer research, which are significant and meet the interests of readers. The information summarized in this manuscript is abundant, but the structure of the manuscript has some inconsistency and needs further improvement.
The manuscript has only two major sections, the "introduction" and the "pancreatic cancer genome". However, the information covered in this review is more than genomic biomarkers of pancreatic cancer. The overall structure of this manuscript should be reorganized.
In the subsection of "2.3. Cancer diagnosis and prognosis by cell-free RNAs", many other circulating targets other than RNA have also been covered, which should be summarized in different subsections.
The information in the part of "2.3.5. ctDNA tests" are overlapping with many points in the previous parts on cfDNAs.
Page 17, the "cccfDNA" is a typo. Few other formatting issues need to be corrected.
Author Response
Reviewer1
The authors of the manuscript “Circulating cell-free nucleic acids as biomarkers for diagnosis and prognosis of pancreatic cancer” appreciate the suggestions. Please find the requested answers below.
1. The manuscript has only two major sections, the "introduction" and the "pancreatic cancer genome". However, the information covered in this review is more than genomic biomarkers of pancreatic cancer. The overall structure of this manuscript should be reorganized:
Answer: We appreciate the suggestions, and we have made some structure alterations, as follows: We kept “Introduction” as section 1, and “Pancreatic cancer genome” as section 2, subdivided into 2.1 Germline context and 2.2 Somatic context. We have created a new section, named “ 3. Pancreatic cancer diagnosis and prognosis by cell-free RNAs”, which was subdivided into 3.1 MicroRNAs (miRNAs); 3.2 Long noncoding RNAs (lncRNAs) and 3.3 Circular RNAs (circRNAs). Another two sections were created: “4. Circulating cell-free tumoral DNA (ctDNA)” and “5. Conclusions”
2. In the subsection of "2.3. Cancer diagnosis and prognosis by cell-free RNAs", many other circulating targets other than RNA have also been covered, which should be summarized in different subsections.
Answer: Structured alterations were made for a more comprehensive text, and DNA context was separated into a new section (section 4).
3. The information in the part of "2.3.5. ctDNA tests" are overlapping with many points in the previous parts on cfDNAs:
Answer: We verified the text and combined the overlapped information into a new one (section 4).
4. Page 17, the "cccfDNA" is a typo. Few other formatting issues need to be corrected.
Answer: A complete screening was made looking for formatting and typo erros.
Reviewer 2 Report
The review article entitled: Circulating cell-free nucleic acids as biomarkers for diagnosis and prognosis of pancreatic cancer has taken into consideration the important diagnostic and therapeutic problem pancreas cancer. The article was well-written, clear, and readable. The importance of the topic derived from the lack of reliable early diagnostic tools. It is important to mention that TC, MRI, or USG have been found useful but in the advanced form of cancer. However, aggressive cancer like the pancreas requires a diagnosis in the early stage. At this point I have some critical remarks, authors should put their attention on the diet risk factors, alcohol, tobacco smoking including electronic or low-temperature cigarettes, also obesity should be considered. Moreover, the DNA damage induction formation and defect in the repair system should be discussed. It would be a wonder if authors describe the metastasis of pancreas cancer too. In conclusion, follow the authors': the review shows the relevance of pancreatic cancer worldwide, giving background on diagnostic complexity and its importance on progression and mortality rate. Therefore I believe that after the answer to my comments, the article will be suitable for publication in the Biomedicines journal.
Author Response
The authors of the manuscript “Circulating cell-free nucleic acids as biomarkers for diagnosis and prognosis of pancreatic cancer” appreciate the suggestions. Please find the requested answers below.
At this point I have some critical remarks, authors should put their attention on the diet risk factors, alcohol, tobacco smoking including electronic or low-temperature cigarettes, also obesity should be considered. Moreover, the DNA damage induction formation and defect in the repair system should be discussed. It would be a wonder if authors describe the metastasis of pancreas cancer too.
Answer: We appreciate the reviewer’s suggestions and inserted a paragraph in the Introduction section describing pancreatic cancer risk factors, which is highlighted in yellow. We also inserted a last paragraph in the Introduction about pancreatic cancer metastasis. However, we considered that insert “DNA damage induction formation and defect in the repair system” may not be necessary in this review, as our main interest is on cell-free nucleic acids and the genetic context of pancreatic cancer was covered in the Introduction.
Reviewer 3 Report
In this manuscript the authors focus on circulating cell-free nucleic acids (cfNA) as biomarkers for diagnosis and prognosis of pancreatic cancer (PC). In introduction epidemiology and current diagnostics of PA were described. Next, germline and somatic context were shown. Finally, microRNAs, long noncoding RNAs, circular RNAs, and circulating cell-free tumoral DNA were presented as the possible biomarkers of detection and prognosis of PA. I like this form of data presentation and feel that it contains a lot of information important for readers. To additionally improve the manuscript I prepared several suggestions.
1. For paragraph 2.2 it will be nice to prepare Figure/Graph to visually summarise somatic molecular alterations known for pancreatic cancer genome, including its functions in tumour development.
2. Because tables should be understood without the main text, please explain under the tables the abbreviations used.
3. Why some references in table 3 are highlighted?
4. Last two paragraphs should be called 4. Summary.
5. In the text there are some editorial mistakes, e.g. p. 2 (Sharma Et al), especificity, p. 17 HIF1A (should be greak alpha).
6. 2.3.4. Circularing Circulating cell-free tumoral DNA (ctDNA).
7. References style must be according to Biomedicines expectations and should appear in the order of appearance with proper number, and not surname and year.
Author Response
The authors of the manuscript “Circulating cell-free nucleic acids as biomarkers for diagnosis and prognosis of pancreatic cancer” appreciate the suggestions. Please find the requested answers below.
1. For paragraph 2.2 it will be nice to prepare Figure/Graph to visually summarise somatic molecular alterations known for pancreatic cancer genome, including its functions in tumour development.
Answer: We appreciate the reviewer suggestion but, despite this indeed is an interesting figure, the somatic alteration on pancreatic cancer genome has been presented as figures in previous reviews (PMID: 34089011; PMID: 33574569; PMID: 30660730).
2. Because tables should be understood without the main text, please explain under the tables the abbreviations used.
Answer: The authors have inserted the following abbreviations in the legend of all tables: “ PCa: Pancreatic cancer; PDAC: Pancreatic ductal adenocarcinoma; AUC: Area under the curve“
3. Why some references in table 3 are highlighted?
Answer: The authors thank this observation. It was a typo error during reference formation and we have properly corrected it.
4. Last two paragraphs should be called 4. Summary.
Answer: The authors thank the suggestion. We have alterated the stucture of the manuscript according to your suggestion, naming the last 2 paragraphs as “5. Conclusions”
5. In the text there are some editorial mistakes, e.g. p. 2 (Sharma Et al), especificity, p. 17 HIF1A (should be greak alpha) AND 6. Circularing Circulatingcell-free tumoral DNA (ctDNA).
Answer: A complete screening was made looking for formatting and typo erros.
7. References style must be according to Biomedicines expectations and should appear in the order of appearance with proper number, and not surname and year.
Answer: References were altered according to Biomedicines journal style.
Round 2
Reviewer 2 Report
The answers to my remarks have been done therefore the article can be accepted for publication.
Reviewer 3 Report
Thank you. I have no additional comments.